# Advances in Chicken Infectious Anemia Vaccines

**DOI:** 10.3390/vaccines13030277

**Published:** 2025-03-05

**Authors:** Haoqian Wang, Yuqing Dan, Linlin Li, Xinwei Wang

**Affiliations:** College of Veterinary Medicine, Henan Agricultural University, No. 218, Ping’an Avenue, Zhengzhou 450046, China; wanghaoqian323@163.com (H.W.); danyq1112@163.com (Y.D.); lilinlinan0409@163.com (L.L.)

**Keywords:** CIAV, etiological, epidemic features, vaccine, narrative review

## Abstract

Chicken infectious anemia (CIA) is caused by the CIA virus (CIAV) and is a globally distributed immunosuppressive disease, resulting in substantial economic losses for the poultry industry. Vaccination is the most cost-effective and efficient strategy for preventing and controlling infectious diseases. The most common CIA vaccines used internationally are attenuated vaccines. Although inactivated vaccines, subunit vaccines, immune complex vaccines, recombinant live viral vector vaccines, and DNA vaccines used for preventing CIAV infection have been developed and exhibited relatively satisfactory immune responses, they have not yet achieved large-scale market applications. Therefore, accelerating the introduction of safe and effective CIA vaccines to the market and developing novel vaccines are crucial for the control of CIA in the poultry industry. This article reviews the etiological characteristics of CIAV, the epidemic features, and the research progress of CIA vaccines, and proposes future research directions, with the aim of providing innovative ideas and scientific references for the research and development of new, safe, and efficient CIA vaccines.

## 1. Introduction

Chicken infectious anemia virus (CIAV) is an immunosuppressive virus, first discovered in Japan in 1979 [1], that causes chicken infectious anemia (CIA), mainly afflicting chicks and inducing aplastic anemia and systemic lymphoid tissue atrophy. This significantly lowers the immunity of chicks, making them highly vulnerable to secondary infections by other viruses, bacteria, and fungi [2]. CIAV can release viral shedding for an extended period through vertical and horizontal transmission. Moreover, live virus vaccines contaminated with CIAV have further aggravated the spread and the difficulty of prevention and control of the epidemic, leading to outbreaks and mixed infections of CIA [3], as well as economic losses in the poultry industry.

Currently, in the poultry industry all over the world, there are no specific treatment approaches for CIA, and only symptomatic treatment can be given to affected chickens. Vaccines are the most effective and economical means for the prevention and control of CIAV infection. At present, the approved CIA vaccines available on the market worldwide mainly consist of attenuated vaccines. It is urgently necessary for the poultry industry to exploit novel CIA vaccines based on the latest technology. This article presents a comprehensive review of the etiological characteristics and epidemic features of CIAV, along with the research progress of its vaccines, aiming to provide certain reference values for the prevention and control of CIAV infection and offer innovative ideas for the research and development of new CIA vaccines.

## 2. Etiology and Epidemic Characteristics of CIAV

CIAV is a single-stranded, negative-strand, circular DNA virus. It has a spherical configuration with a diameter of approximately 17 nanometers, consists of 32 capsomeres, and the nucleocapsid has an icosahedral symmetry structure, which endows it with unique stability. Moreover, it is the only member of the genus *Gyrovirus* in the family *Anelloviridae*. The whole genome of CIAV is about 2.3 kb, containing open reading frames (ORFs) encoding three proteins (VP1, VP2, and VP3) [2]. VP1 is not only the structural protein (capsid protein) but also the main immunogenic protein of the virus. It has a molecular weight of 51.2 K Da and is capable of stimulating the host to produce antibodies [4]. VP2, a non-structural protein with a molecular weight of 24 K Da, serves as a scaffold protein during virus assembly and can assist the VP1 protein in attaining the correct conformation and exposing its epitopes [5]. VP3, also referred to as apoptin, with a molecular weight of 13.6 k Da, is capable of inducing the apoptosis of infected cells [6,7].

The main targets of CIAV are the blood cells in the bone marrow and the precursor lymphocytes in the thymus [8,9]. Once chicks are infected with CIAV, on the one hand, the hematopoietic cells in their bone marrow are impaired, with their proliferation and differentiation capabilities inhibited. This disrupts the homeostasis of the hematopoietic microenvironment in the bone marrow, eventually leading to a reduction in the number of red blood cells and myeloid cells in the diseased chickens, triggering aplastic anemia and presenting clinical symptoms like anemia [10]. On the other hand, CIAV can damage the cortical lymphocytes in the thymus. The T-lymphocyte precursor cells among them are relatively sensitive to CIAV. The precursor lymphocytes are the precursor stage of the T cells that will exert immune functions. The virus replicates abundantly in the T-cell precursors in the thymus cortex. The VP3 of CIAV can induce apoptosis of the precursor lymphocytes, causing cell death and a decrease in the number of precursor lymphocytes in the thymus. This affects the subsequent generation of T lymphocytes, further reducing the cellular immune function of chickens and their resistance to pathogens. Consequently, this makes them prone to secondary or concurrent infections with other pathogens, resulting in an increased mortality and culling rate of the chicken flock [11].

CIAV can be transmitted horizontally and vertically [3]. The horizontal transmission routes are diverse, including contact with contaminants such as the feces and blood of infected chickens, being transmitted to other chickens in the same flock through respiratory and digestive tracts, etc. Vertical transmission occurs when breeder chickens pass the virus to the next generation of chicks. This is the main infection route and poses a serious threat to the health of chicks. Under natural conditions, CIAV infection exhibits distinct age-related characteristics, and chicks at 1–3 weeks of age are highly susceptible to it. Adult chickens typically exhibit subclinical infections [12,13], and the virus can persist in their bodies for a long time and continuously shed to the outside, further intensifying the spread of the epidemic and the difficulty of prevention and control, complicating eradication efforts in commercial flocks [14]. CIAV frequently co-infects with other viral agents, which can often bring about alterations in the biological characteristics of both the virus and the host cells [15,16]. This co-infection not only diminishes the efficacy of certain vaccines but also strengthens the replication capacity of other viruses, resulting in a synergistic increase in viral pathogenicity and immunosuppression [17,18,19,20,21,22]. Furthermore, the use of live virus vaccines contaminated with CIAV is also one of the reasons for the large-scale outbreaks and mixed infections of CIA [3].

The mortality among chickens due to CIAV-infection is affected by multiple clinical factors. For chicks, the mortality may be as high as 50% [23], while for adult chickens it is usually extremely low or even zero. Notably, the incidence of CIA is usually high in some countries. According to relevant research reports, from 2017 to 2020, Li et al. used multiplex reverse-transcription quantitative real-time PCR (MRT-qPCR) to test six major diseases in 1187 poultry samples of different ages and breeds from China. It was found that, compared with Marek’s disease virus (MDV), reticuloendotheliosis virus (REV), avian reovirus (ARV), infectious bursal disease virus (IBDV), and fowl adenovirus (FAdV), the detection rate of the nucleic acid of CIAV was the highest [24]. The epidemiological investigation results of CIAV in China during the period from 2020 to 2022 indicate that the positive rate of CIAV DNA in chicken flocks from each province ranged from 50% to 80% [25,26]. Among a total of 119 samples from 64 farms in northern Vietnam, it was discovered that in 47 (73.4%) of the farms, a total of 74 (62.2%) samples tested positive for CIAV by PCR [27]. Bhatt et al. randomly collected a total of 404 serum samples from broilers, laying hens, and breeding hens of various age groups (ranging from 3 days to 52 weeks) from 11 poultry farms in India for an epidemiological investigation of CIAV. It was discovered that the positive rate of CIAV antibodies reached 86.88% [28]. Due to the frequent economic connections among different countries and regions, CIAV has spread globally [29], including to China [30], South Korea [31], Japan [32], Nigeria [33], Vietnam [34], Egypt [35], and Finland [36], and has given rise to significant economic losses in the global poultry breeding industry. Consequently, it is imperative to explore new strategies for the prevention and control of CIA.

## 3. Current Status of Vaccine Research

### 3.1. Attenuated Live Vaccines

Attenuated vaccines are a type of vaccine in which the pathogen is attenuated in virulence while still retaining excellent antigenicity. These are the preferred vaccines in the global veterinary vaccine market and possess the characteristics of a strong immune response effect, a long duration, fewer vaccinations required, a smaller dosage, and a relatively low cost. Attenuated vaccines accounted for 24.3% of the global market share by 2022 [37]. For attenuated CIA vaccines, to date they have been extensively employed in breeder flocks to control CIA in offspring chicken.

Some relevant scholars have conducted evaluations of the attenuated CIA vaccines. They inoculated 1-day-old chicks with both the pathogenic CIAV strain (Cux-1) and the attenuated CIAV strains (CI 34 and CRI 18). The findings revealed that the attenuated CIAV strains significantly diminished the pathogenic effects on 1-day-old chicks [38]. At the same time, they also demonstrated that vaccinating 1-day-old chicks with the attenuated CIA vaccine is not the most preferable vaccination approach. Other scholars passaged the attenuated Cux-1 strain through cell culture 173 times to obtain the attenuated strain “cloned isolate 10” to develop an attenuated CIA vaccine and inoculate chickens aged 9–15 weeks. It was determined that this vaccine was non-pathogenic for hens within the age range of 9–15 weeks, and it effectively prevented the onset of CIA in their progeny [39]. In order to verify the viral persistence of the attenuated CIA vaccine and its related lymphocytic diseases, Vaziry et al. used the commercial CIA vaccine (CIAV-VAC) to vaccinate 1-day-old chicks. The results indicated that the chicks exhibited subclinical infection symptoms, with the antibody levels in most vaccinated individuals being notably low and not exceeding a duration of 18 days. Furthermore, the CIAV-VAC strain persisted in the spleens and thymuses of the chicks and induced thymic lymphocytic diseases [40]. During the exploration of the immune pathogenesis of the attenuated CIA vaccine, the same result was also found, which showed that the attenuated CIA vaccine strain is not suitable for the vaccination of one-day-old chickens [41]. As a result, the attenuated CIA vaccines that have received authorization for use in the United States are exclusively recommended for breeder chickens between the ages of 9 and 15 weeks [42]. This approach enables the protection of progeny chicks against CIAV through maternal antibodies [43].

Currently, there are various commercialized attenuated vaccines against CIAV infection (Table 1). Although commercial CIA vaccines can offer excellent immune protection to chicks after being administered to breeding chickens, the CIA vaccine strain can persist in some immune chicks and cause thymic lymphoid cell disorders [40], and the attenuated strain of CIAV has the phenomenon of virulence reversion, which is a disadvantage of its use [39].

### 3.2. Inactivated Killed Vaccines

Inactivated vaccines are defined as vaccines in which pathogenic microorganisms are killed through physical or chemical methods, causing them to lose their infectivity and multiplication activity while retaining their immunogenicity, and then combined with the corresponding adjuvants. Inactivated vaccines have the advantages of high safety, stable properties, easy storage, and not being prone to causing pollution. Pages-Mante et al. evaluated the inactivated CIA vaccine and found that the offspring of hens vaccinated with it still had sufficient levels of maternal antibodies to CIAV at 3–4 weeks of age, demonstrating that the inactivated CIA vaccine is an effective measure for the prevention and control of CIAV [44]. To optimize the production process of inactivated vaccines, some researchers have developed an inactivated CIA vaccine by utilizing β-propiolactone to inactivate the GD-G-12 CIAV strain, combined it with oil-based adjuvants, and immunized 13-week-old breeder chickens. This modified inactivated CIA vaccine was shown to offer vaccine protection rates of 98 to 100% for chickens [14]. Compared with the traditional formaldehyde inactivation method, the hydrolysis of β-propiolactone provided a more complete and rapid inactivation of the virus [45]. Although inactivated CIA vaccines can protect chickens from CIAV infection, compared with attenuated vaccines, they demand a large dosage, and the immune protection that they elicit has certain limitations. For instance, inactivated CIA vaccines predominantly introduce humoral immunity and limited cellular immunity, while attenuated CIA vaccines are capable of triggering long-lasting cellular immunity and humoral immunity. Furthermore, the viral load of CIAV in cells or chicken embryos is relatively low, giving rise to a high production cost for inactivated CIA vaccines. These issues have precluded the extensive commercial utilization of inactivated CIA vaccines [14].

### 3.3. Subunit Vaccines

Subunit vaccines typically denote those prepared by obtaining the antigenic components of microbial pathogens via genetic engineering. These vaccines do not incorporate any side-effect material and are capable of eliciting an immune response within the host organism. Compared to inactivated vaccines and attenuated vaccines, subunit vaccines do not possess the entire structure of the virus, and they are safer in clinical applications. However, due to the weak immunogenicity of these vaccines, adjuvants are frequently added to them to boost their immune effect. Moreover, further safety assessment of some adjuvants may need to be performed.

Shen et al. utilized the *Escherichia coli* expression system to construct and express a recombinant subunit vaccine consisting of CIAV VP1 and rPiIFN-γ. In comparative analyses, the rVP1 + recombinant pigeon INF-γ (rPiINF-γ) vaccine was developed to evoke significantly higher antibody titers and Th-1-type cytokines than both the VP1 subunit vaccine and the inactivated CIA vaccine, thereby prominently enhancing both the humoral and cellular immune responses. This discovery indicates that rPiIFN-γ acts effectively as an adjuvant, enhancing the immunogenicity of the vaccine [46]. At the same time, it was also found that, although there are numerous neutralizing epitopes in VP1, the immunogenicity generated by the VP1 protein alone is weak [2,47]. Another investigation utilized the *Escherichia coli* system to express VP1, VP2, and VP3 individually, revealing that the combination of VP1 and VP2 yielded higher antibody titers compared to the individual expression of VP1, VP2, and VP3, thus resulting in enhanced protection. Not only that, the author also compared Freund’s adjuvants with CpG-ODN and found that both adjuvants can increase the immune response. However, CpG-ODN can better fortify the cellular and humoral immunity of the CIA subunit vaccine than Freund’s adjuvant, achieving a superior immune response outcome [48]. Tseng et al. developed a recombinant baculovirus expression system that is capable of expressing the CIAV *VP1*, *VP2*, and *Chicken IL-12 (chIL-12)* genes. Among them, recombinant VP1 can produce self-assembled virus-like particles (VLPs), and chIL-12 can act as an adjuvant to strengthen the immune response of vaccinated chickens and induce higher antibody titers than commercial vaccines. In summary, subunit vaccines may provide an effective vaccination design strategy without the safety concerns of the attenuated live CIA vaccine [49]. rPiIFN-γ, Freund’s adjuvant, CpG-ODN, and chIL-12, when used as adjuvants for CIA subunit vaccines, can significantly improve the immune effect of the vaccines. Therefore, the selection of a suitable adjuvant or the development of novel adjuvants for CIA subunit vaccines is important for protecting chicks against CIAV infection. Although subunit vaccines show good performance in terms of safety, unfortunately, there is no mature commercialized product available at present.

### 3.4. Immune Complex Vaccines

Immune complex vaccines are a type of vaccine formed by combining antigens or immunogens with antibodies to form immune complexes [50]. These vaccines present several advantages, including a reduction in the frequency of vaccinations required, enhancement of the vaccine’s immunogenicity, facilitation of generating more robust and sustained immune responses, and better safety in use. Schat et al. prepared immune complexes (ICP80 and ICP160) containing specific neutralizing units of CIAV. They inoculated 1-day-old chicks without maternal antibodies and challenged them with the CIAV strain (01-420) at two or three weeks of age. Studies have shown that immune complex vaccines significantly delay viral replication and, to some extent, alleviate the clinical symptoms of chicken infectious anemia virus infection. Consequently, immune complex vaccines may offer a viable strategy for safeguarding newly hatched chicks from the virus in the field [51]. However, for reasons unknown, immune complex vaccines remain uncommercialized at present.

### 3.5. Live Viral Vector Vaccines

Recombinant viral vector vaccines are a type of vaccine that uses a virus as a vector to deliver the target antigen. These vaccines are capable of mimicking the natural infection process, thereby eliciting a robust immune response and establishing long-term immunological memory. Therefore, recombinant viral vector vaccines have become an important approach in vaccine research and development, and they have also promoted the development of live viral vector vaccines in the CIAV field.

Chellappa et al. employed the Newcastle Disease Virus (NDV) strain R2B as a vector to construct an NDV vector vaccine that concurrently expresses the *VP1* and *VP2* genes of CIAV based on reverse genetics technology. This vaccine induced strong humoral and cellular immune responses, offering protection for one-week-old chickens [52]. Additionally, Li et al. developed a CIA vaccine based on the MDV viral vector by integrating the CIAV VP1 and VP2 genes into the US2 locus of the MDV vaccine strain 814 via a fosmid-based rescue system and subsequently vaccinating one-day-old chicks. The results indicated that the rMDV-CIAV-2 vaccine can trigger a high antibody titer against CIAV and exert a good immune-protective effect [53]. However, recombinant viral vector vaccines have the problem of antibody persistence. Further studies are needed to analyze the immune duration of the recombinant viruses and optimize the vaccination doses to better prevent CIAV infection. Live viral vector vaccines constitute one of the promising vaccine varieties. Likewise, no commercialized products are available at present.

### 3.6. Nucleic Acid Vaccines

DNA vaccines are a type of vaccine in which exogenous genes encoding a certain antigen protein are inserted into eukaryotic expression vectors, and then the recombinant plasmids are directly introduced into the animal’s body, allowing the exogenous genes to be expressed in the host cells, thereby inducing the immune response of the body. DNA vaccines are safe, stable, easy to store, and can induce humoral and cellular immunity [2]. However, it has been noted that the majority of DNA vaccines developed demonstrate lower immunogenicity, resulting in a limited number of commercialized DNA vaccines [54].

Moeini et al. made use of the eukaryotic co-expression vector pBudCE4.1 to construct the pBudVP2-VP1 DNA vaccine. This vaccine can effectively stimulate the body to exert cellular and humoral immune responses [55]. In the same year, their research team once again used the pBudCE4.1 vector to form a fusion of the *VP1* and *VP2* genes of CIAV with the *VP22* gene from MDV, leading to the construction of the pBudVP2-VP1/VP22 DNA vaccine. Immunological assessments carried out via intramuscular injection showed that the group receiving the pBudVP2-VP1/VP22 vaccine presented significantly higher titers of CIAV antibodies, a greater number of splenic cells, and higher levels of Th1 cytokines compared to the group vaccinated with pBudVP2-VP1 alone. These results suggest that the integration of VP22 into CIAV VP2-VP1 increased the efficacy of the DNA vaccine against CIAV [56].

Research has shown that the co-immunization of high-mobility group box 1 protein (HMGB1) as an adjuvant can trigger strong cellular and humoral immune responses [57,58]. In light of this, Sawant et al. have developed a DNA vaccine expressing the CIAV *VP1* and *VP2* genes for combined use with truncated chicken high-mobility group box 1 (HMGB1ΔC) protein. Co-immunization studies carried out in chicks disclosed that HMGB1ΔC significantly increased both the cellular and humoral immune responses evoked by the CIA DNA vaccine, thus offering considerable protection for the chicks and underlining the potential of HMGB1ΔC as an efficient immune adjuvant for CIA DNA vaccination [54]. HMGB1ΔC has shown promising protective immune responses as a novel immune adjuvant, although it may have deficiencies in terms of immune persistence. This could potentially be attributed to the limited expression duration of the DNA vaccine within the body, or to the development of tolerance to the vaccine components by the immune systems of the animals. Although adjuvants can markedly improve the immune protective efficacy of DNA vaccines, how to guarantee the stability and efficient transfection of DNA vaccines in vivo remains a technical challenge. These drawbacks constrain the commercialization of this type of vaccine.

## 4. Conclusions

This article reviews the current etiological characteristics, epidemic features, and research progress of CIAV vaccines. CIAV is globally prevalent. The positive rate of CIAV remains exceedingly high in some regions, causing considerable economic losses to the poultry industry. Vaccination is the chief measure for preventing CIAV infection. At present, attenuated vaccines are the main CIA vaccines utilized internationally. It is crucial to note that CIAV mainly impairs chicks, while commercialized attenuated live CIA vaccines are primarily used for the immunization of breeder chickens and pose a risk of viral reversion to its ancestral state. The CIA immune complex vaccine can be used in ovo or in newly hatched chickens, and it has shown good efficacy in inhibiting viral replication. The inactivated CIA vaccine is characterized by high safety. Nevertheless, the inoculation dosage of the inactivated CIA vaccine is large, and proliferation of the CIAV virus in the vaccine production process is relatively difficult, which leads to a high cost of the inactivated CIA vaccine. It also limits the market approval of inactivated CIA vaccines to a certain extent. Therefore, optimizing the production process to increase the viral titers of CIAV is of crucial significance for promoting the large-scale application of inactivated CIA vaccines. The characteristics of subunit vaccines are similar to those of inactivated vaccines. However, the immunogenicity of subunit vaccines is relatively weak. Therefore, the combined use of CIA subunit vaccines with adjuvants, such as rPiIFN-γ, Freund’s adjuvant, CpG-ODN, and chIL-12, may be a strategy to develop CIA vaccines. DNA vaccines can enhance the immune response to CIAV when used in conjunction with adjuvants, and they hold great potential in the prevention and control of CIA. The strategy of combining DNA vaccines and subunit vaccines [11] can induce higher levels of antibody responses and cellular immune responses in chickens [2], and this strategy is likely to potentially become an effective means for the prevention and control of CIA in the future. Additionally, CIA live viral vector vaccines may be worth developing. However, except for the attenuated live CIA vaccine, other types of CIA vaccines have not yet been approved for large-scale production [59]. This situation accentuates the urgent market demand for CIA vaccines that are safe, effective, and capable of providing robust immunoprotection.

## 5. Future Directions

To effectively prevent and control CIA infection, surmount the challenges encountered in the current application of CIA vaccines, and meet the market’s demands, future research could be carried out in the following directions:

In terms of vaccine strain optimization, there is a need for further screening, cultivation, or modification to obtain safer and more efficient vaccine strains. Through the monitoring and analysis of prevalent CIAV strains, targeted vaccine strains could be developed to enhance the prevention and control efficacy of the vaccines. Meanwhile, genetic engineering techniques could be employed to modify the existing vaccine strains, such as optimizing the viral gene sequence to enhance the stability and immunogenicity of the vaccine strains.

The development of new vaccine vector systems and adjuvants constitutes a crucial research direction in the future. Novel viral and bacterial vectors should be explored to seek vector systems with superior safety and stability. Simultaneously, the research on novel adjuvants, such as nano-adjuvants and immune-stimulating complexes, could be strengthened to enhance the immune efficacy of vaccines, while decreasing their dosage and vaccination frequency. The synergy mechanisms between different adjuvants and vaccine antigens could be studied to optimize the adjuvant formulations and develop new adjuvants more appropriate for CIA vaccines.

The application of novel vaccine technologies is of paramount significance for expediting the development of safe and effective new CIA vaccines. mRNA vaccines have demonstrated remarkable efficacy in the prevention and control of COVID-19 [60]. In contrast to conventional vaccines, mRNA vaccines boast a shorter research and development period, a simplified production process, and relatively good safety and antigenicity (capable of eliciting potent cellular and humoral immune responses). However, it is important to note that mRNA vaccines have stringent requirements for storage and transportation. Overall, mRNA vaccine technology has shown great potential in virus prevention and control. Probiotic oral vaccines represent a novel type of vaccine technology that utilizes probiotics as carriers for antigen delivery, thereby inducing the body to generate an immune response [61]. Compared with traditional injectable vaccines, the administration method of oral vaccines is more convenient. Probiotics are not only beneficial microorganisms to the body, capable of regulating the balance of the intestinal microecology and improving the intestinal immune environment, but can also act as vaccine adjuvants to enhance the mucosal immune response of the body and play an important role in the first line of pathogen invasion. Currently, neither of these two new vaccine technologies has been applied in the research and development of CIA vaccines. Computational vaccinology, an emerging and promising discipline, primarily employs computer technology and bioinformatics techniques to assist researchers in rapidly identifying potential antigenic epitopes, thereby expediting the research and development processes for novel vaccines targeting infectious diseases [62,63]. Therefore, using computational vaccinology to guide the development of new types of CIA vaccines, such as mRNA and probiotic vaccines, would be a new strategy and be worthy of further exploration. Synthetic biology, an emerging interdisciplinary discipline, possesses the capability to develop safe and efficient synthetic viral vaccines within a relatively short timeframe [64]. In the future, this field is expected to provide new ideas and methods for dealing with CIAV infection.

## 6. Limitations

We conducted a narrative review of CIA vaccines and also shared our insights. However, it may be possible that some relevant studies were not completely utilized or quoted in this article, for it is not a systematic review.

## Figures and Tables

**Table 1 vaccines-13-00277-t001:** Commercialized attenuated CIA vaccines.

Name	Company Name	Strain Name	Storage Temperature	Vaccine Status	Mode of Administration	Vaccination Target	Country
CIRCOMUNE^®^	Ceva Santé Animale, Libourne	Del Ros	2–7 °C	Liquid	Use only by wing-webadministration	Chickens at the age of 9 to 12 weeks	France
Nobilis CAV P4	MSD Animal Health, Madison	26P4	2–8 °C	Freeze-dried granules	Intramuscular orsubcutaneous injection	Starting from 8 weeks of age until 6 weeks before the start of laying	USA
AviPro THYMOVAC Lyophilisate	Lohmann Animal Health, Cuxhaven	Cux-1	2–8 °C	Freeze-dried granules	Drinking water	Starting from 8 weeks of age until 6 weeks before the start of laying	Germany
CAV-VAC^®^	Merck Animal Health, Madison	A modified U.S. field isolate	2–8 °C	Liquid	Vaccine is applied to the web of the wing	For chickens 10–12 weeks of age	USA
Gyrovac	BIOVAC, or Akiva	OA1	2–8 °C	Freeze-dried granules	Drinking water	Healthy breeding chickens or laying hens	Israel

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
