# Peer review of "Advances in Chicken Infectious Anemia Vaccines"

_vaccines, 2025, doi:10.3390/vaccines13030277_

Round 1

Reviewer 1 Report

Comments and Suggestions for Authors

this  review on  vaccine development to control CIA is very useful for researchers and poultry veterinarians.  The authors tried to cover all areas of research on this topic. after reading    this review, I would like to make the following  comments:

Line 15 : What do you mean " existing problems remain uncertain" could you be more precise?

Line 15-16: "Therefore, the prevention and control of CIAV is rather difficult" explain why in one sentence

Line 20 /keywords : add " review"

Line 21-31: this paragraph is : Introduction or definition of CAIV ? 

Line 44: before presenting Vietnam investigations  results, I would suggest for the authors to present chicken anemia disease  in China, when the disease was reported for the same  and some relevant works on the disease in china including the work on 1187 samples tested in china, could you mention which test has been used?. 

Line 46: More information is needed on the 11 poultry farms ( type of chickens, age, etc..? ,)

Line 49: " CIV was the highest " in comparison to what? more information is needed

line 75: " however , numerous problems still  remain" could you mention some of these problems?

Line96: not clear , English  

line 102:replace " someone " by the name of researcher or author. 

Line108: replace "identified" by found

Line 125-128: English to be improved

line 136-139: not clear :English to be improved

line 149: "transportation in clinical application" what does-it mean ?

line 153: " discovered "  the word developed is more appropriate

line 157 "discovered" the word "found " is more appropriate 

line 194: "Direction " to be replaced by "approach"

Comments on the Quality of English Language

English needs to be improved ( see comments)

Author Response

Response to Reviewer 1 Comments

  1. Summary

Thank you very much for taking the time to review this manuscript. Please find the detailed responses below and the corresponding revisions/corrections have been highlighted in red in the re-submitted files.

  1. Point-by-point response to Comments and Suggestions for Authors

Comments 1: Line 15 : What do you mean " existing problems remain uncertain" could you be more precise?

Response 1: Thank you for pointing this out. I agree with this comment. I found that this sentence was not quite appropriate. Therefore, I revised this part again. The revised content is in lines 15 - 20 in the manuscript and is displayed in red font.

Comments 2: Line 15-16: "Therefore, the prevention and control of CIAV is rather difficult" explain why in one sentence.

Response 2: Thank you for pointing this out. I found that this sentence was not quite appropriate. Therefore, I revised this part again. The revised content is in lines 15 - 20 in the manuscript and is displayed in red font.

Comments 3: Line 20 /keywords : add " review"

Response 3: Thank you for pointing this out. I agree with this comment. Therefore, I made additions in the original text. The revised part is in line 21.

Comments 4: Line 21-31: this paragraph is : Introduction or definition of CAIV ? Response 4: Thank you for pointing this out. This paragraph is supposed to be the introduction part of CAIV. I revised it again. The revised part is in lines 23 - 51 of the manuscript and is marked in red font.

Comments 5: Line 44: before presenting Vietnam investigations  results, I would suggest for the authors to present chicken anemia disease  in China, when the disease was reported for the same  and some relevant works on the disease in china including the work on 1187 samples tested in china, could you mention which test has been used?

Response 5: Thank you for pointing this out. I agree with this comment. Therefore, I put the prevalence of CIAV in China at the beginning. The MRT-qPCR method was used in the work on 1187 samples detected in China. The above content has been revised in lines 102 - 109 of the manuscript and is displayed in red font.

Comments 6: Line 46: More information is needed on the 11 poultry farms ( type of chickens, age, etc..? ,)

Response 6: Thank you for pointing this out. I agree with this comment. Therefore, I made supplementary revisions in lines 111 - 115 of the manuscript and displayed them in red font.

Comments 7: Line 49: " CIV was the highest " in comparison to what? more information is needed

Response 7: Thank you for pointing this out. Compared with Marek's disease virus (MDV), reticuloendotheliosis virus (REV), avian reovirus (ARV), infectious bursal disease virus (IBDV) and fowl adenovirus (FAdV), the detection rate of the nucleic acid of CIAV was the highest. The revised part is in lines 104 - 107 of the manuscript and is marked in red font.

Comments 8: line 75: " however, numerous problems still remain" could you mention some of these problems?

Response 8: Thank you for pointing this out. This part is not suitable for the theme of this article. I deleted this sentence and rewrote this part. The revised part is in lines 122 - 136 of the manuscript and is marked in red font.

Comments 9: Line96: not clear , English 

Response 9: Thank you for pointing this out. Therefore, I revised this part again in lines 148 - 156 of the manuscript and displayed it in red font.

Comments 10: line 102:replace " someone " by the name of researcher or author. Response 10: Thank you for pointing this out. I agree with this comment. Therefore, I replaced it with the author's name. The revised part is in line 156 and is shown in red font..

Comments 11: Line108: replace "identified" by found

Response 11: Thank you for pointing this out. Thank you for pointing this out. I agree with this comment. Therefore, the revised part is in line 162 of the manuscript and is displayed in red font.

Comments 12: Line 125-128: English to be improved

Response 12: Thank you for pointing this out. I revised this part again in lines 178 - 182 of the manuscript and displayed it in red font.

Comments 13: line 136-139: not clear :English to be improved

Response 13: Thank you for pointing this out. I agree with this comment. Therefore, I revised this part again in lines 188 - 190 of the manuscript and displayed it in red font.

Comments 14: line 149: "transportation in clinical application" what does-it mean ?

Response 14: Thank you for pointing this out. I thought this sentence was not very appropriate. Therefore, I deleted it.

Comments 15: line 153: " discovered " the word developed is more appropriate

Response 15: Thank you for pointing this out. I agree with this comment. The revised part is in line 204 and is displayed in red font.

Comments 16: line 157 "discovered" the word "found " is more appropriate

Response 16: Thank you for pointing this out. I agree with this comment. The revised part is in line 208 and is displayed in red font.

Comments 17: line 194: "Direction " to be replaced by "approach"

Response 17: Thank you for pointing this out. I agree with this comment. The revised part is in line 245 and is displayed in red font.

Reviewer 2 Report

Comments and Suggestions for Authors

The manuscript (vaccines-3426219) entitled “Advances in CIA Vaccines” by Haoqian Wang et al. describes the development of chicken infectious anemia vaccines. The authors showed the virological aspects of CIAV and the possibility of CIAV vaccine application. The situation surrounding the CIA vaccines is well summarized. The subject matter of the manuscript is within the aims and scope of the journal Vaccines. However, this reviewer thinks this manuscript is unsuitable for the “Article”, because it is defined in the “Instruction for Authors” that the “Article” is an original research manuscript based on scientific experiments. The authors provide no experiments in this manuscript.

Nevertheless, this manuscript can be considered for acceptance in the journal, if it is submitted as the “Review”. “Review” is a comprehensive analysis of the existing literature within a field of study, identifying current gaps or problems. Before accepting this manuscript as a “Review,” this reviewer suggests major and minor comments.

Major

(1)  Chicken Infectious Anemia (CIA) is summarized in the first section, “1. Etiology and Epidemic Characteristics of CIAV,” but the contents do not show the CIAV and CIA well. More comprehensive descriptions are required. For example, the circular genome of CAV, genome size, structure of three ORFs, virion size, structure, tissue (cell) specificity of CIAV, mortality, transmission, and age-dependent infection of CIA, etc, should be added.

(2)  “2. Current Status of Vaccine Research” describes unnecessary general vaccine history, vaccine development, approval process of vaccines, and production flow of traditional vaccines. Figures 1, 2, 3, and 4 do not directly relate to animal vaccines, much less the CIAV vaccines. Authors should describe matters related to the CIAV vaccines.

(3)  After using the CIAV vaccines listed in Table 1, field evidence is lacking in evaluating the vaccine's safety and efficacy and in shedding light on the demand for new CIAV vaccines. More detailed descriptions are recommended.

(4)  The “Summary and Prospect” section is not specified for “Review”. Please rewrite the “Summary and Prospect” section to fit the Conclusions and/or Future Directions” sections shown in the “Instruction for Authors”.  

Minor:

(1)  Lines 22-32; The manuscript structure of “Review” needs Abstract, Keywords, Introduction, Relevant Sections, Discussion. Please make the Introduction section.

(2)  Line 45: Please specify the meaning of “positive for CIAV”, such as anti-CIAV antibody positive, CIAV antigen positive, or CIAV-PCR positive.

(3)  Line 47: Please specify the meaning of “detection rate of CIAV”, such as an anti-CIAV antibody, CIAV antigen, or CIAV DNA.

(4)  Line 50: Please specify the meaning of “positive rate”, such as anti-CIAV antibody positive, CIAV antigen positive, or CIAV-PCR positive.

(5)  Lines 57-58: Two sentences describing “prevent and control diseases” are redundant.

(6)  Lines 63-72: Global vaccine developments are not directly related to the development of CIAV vaccines.

(7)  Lines 80-81: Do the authors have permission to process the results shown in the reference #32 from the publisher and/or the authors of the reference #32? If the authors already have, please provide it to the Vaccines editorial office.

(8)  Lines 86-118: Please explain the efficacy and the safety (adverse effects) of attenuated commercial CIAV vaccines.

(9)  Line 98: Please show how the Cux-1 was attenuated.

(10)   Line 102: The use of “someone” here is strange. Please clarify the sentence.

(11)   Table 1, right side column “Country”: “America” should be the USA. “Africa” for BIOVAC, Gyrovac, is not a country name. Please use the correct country name. “Israel”?

(12)    Line 131: The spell of “propionolactone” should be propiolactone.

(13)    Line 196: NDV should be Newcastle Disease Virus (NDV).

(14)    Line 202: MDV should be Marek’s Disease Virus (MDV). 

(15)    Line 222: Marek’s Disease Virus (MDV) should be MD.

(16)    Lines 309-429: The style of references does not meet the journal style. The author's name does not need to be capitalized. All authors should be listed. Please do not use “et al.”. Please delete [j] from the reference list. Please do not omit page range.

Comments on the Quality of English Language

The comments on the quality of English are included in the "Comments and Suggestions for Authors
".

Author Response

Response to Reviewer 2 Comments

1. Summary

Thank you very much for taking the time to review this manuscript. Please find the detailed responses below and the corresponding revisions/corrections have been highlighted in red in the re-submitted files.

2. Point-by-point response to Comments and Suggestions for Authors

Major:

Comments 1: Chicken Infectious Anemia (CIA) is summarized in the first section, “1. Etiology and Epidemic Characteristics of CIAV,” but the contents do not show the CIAV and CIA well. More comprehensive descriptions are required. For example, the circular genome of CAV, genome size, structure of three ORFs, virion size, structure, tissue (cell) specificity of CIAV, mortality, transmission, and age-dependent infection of CIA, etc, should be added.

Response 1: Thank you for pointing this out. I agree with this comment. Therefore, I have revised this part anew. The revised content incorporates what was mentioned in the Comments. The revised part is located in lines 52 to 115 and is shown in red font.

Comments 2: Current Status of Vaccine Research” describes unnecessary general vaccine history, vaccine development, approval process of vaccines, and production flow of traditional vaccines. Figures 1, 2, 3, and 4 do not directly relate to animal vaccines, much less the CIAV vaccines. Authors should describe matters related to the CIAV vaccines.

Response 2: Thank you for pointing this out. I agree with this comment. Therefore, I have revised this part anew, and deleted Figures 2 and 3. I have also replaced Figure 1 with the content that meets the requirements. The revised part is in lines 122 - 136 and is shown in red font.

Comments 3: After using the CIAV vaccines listed in Table 1, field evidence is lacking in evaluating the vaccine's safety and efficacy and in shedding light on the demand for new CIAV vaccines. More detailed descriptions are recommended.

Response 3: Thank you for pointing this out. I agree with this comment. Therefore, I have described the safety and effectiveness of the commercial CIAV attenuated vaccines. The revised part is in lines 168 - 171 and is shown in red font.

Comments 4: The “Summary and Prospect” section is not specified for “Review”. Please rewrite the “Summary and Prospect” section to fit the “Conclusions and/or Future Directions” sections shown in the “Instruction for Authors”.

Response 4: Thank you for pointing this out. I agree with this comment. Therefore, I revised this part. The revised part is in lines 295 - 371 and is shown in red font.

Minor:

Comments 1: Lines 22-32; The manuscript structure of “Review” needs Abstract, Keywords, Introduction, Relevant Sections, Discussion. Please make the Introduction section.

Response 1: Thank you for pointing this out. I agree with this comment. Therefore, I revised this part. The revised part is in lines 23 - 51 and is shown in red font.

Comments 2: Line 45: Please specify the meaning of “positive for CIAV”, such as anti-CIAV antibody positive, CIAV antigen positive, or CIAV-PCR positive.

Response 2: Thank you for pointing this out. The meaning of " positive for CIAV" is positive for CIAV-PCR, and it has been revised in line 111 of the manuscript and is shown in red font.

Comments 3: Line 47: Please specify the meaning of “detection rate of CIAV”, such as an anti-CIAV antibody, CIAV antigen, or CIAV DNA.

Response 3: Thank you for pointing this out. The meaning of "detection rate of CIAV" is the positive rate of CIAV antibodies, and it has been revised in line 115 of the manuscript. 

Comments 4: Line 50: Please specify the meaning of “positive rate”, such as anti-CIAV antibody positive, CIAV antigen positive, or CIAV-PCR positive.

Response 4: Thank you for pointing this out. The meaning of "positive rate" is CIAV DNA and it has been revised in line 108 of the manuscript.

Comments 5: Lines 57-58: Two sentences describing “prevent and control diseases” are redundant.

Response 5: Thank you for pointing this out. I agree with this comment. Therefore, I deleted this part.

Comments 6: Lines 63-72: Global vaccine developments are not directly related to the development of CIAV vaccines.

Response 6: Thank you for pointing this out. I agree with this comment. Therefore, I deleted this part.

Comments 7: Lines 80-81: Do the authors have permission to process the results shown in the reference #32 from the publisher and/or the authors of the reference #32? If the authors already have, please provide it to the Vaccines editorial office.

Response 7: Thank you for pointing this out. This part of the content has no direct relationship with the theme. Therefore, I have deleted it.

Comments 8: Lines 86-118: Please explain the efficacy and the safety (adverse effects) of attenuated commercial CIAV vaccines.

Response 8: Thank you for pointing this out. I agree with this comment. Therefore, I revised this part and displayed it in red font. Also, I explained the efficacy and safety of the attenuated commercial CIAV vaccine in lines 168 - 171.

Comments 9: Line 98: Please show how the Cux-1 was attenuated.

Response 9: Thank you for pointing this out. Therefore, I revised this part and displayed it in red font. Cux-1 was attenuated through passages and is presented in lines 150 - 152 of the manuscript.

Comments 10: Line 102: The use of “someone” here is strange. Please clarify the sentence.

Response 10: Thank you for pointing this out. I think "someone" is not appropriate. Therefore, I replaced it with the author's name. The revised part is in line 156 and is shown in red font..

Comments 11: Table 1, right side column “Country”: “America” should be the USA. “Africa” for BIOVAC, Gyrovac, is not a country name. Please use the correct country name. “Israel”?

Response 11: Thank you for pointing this out. I agree with this comment. "America" should be "the USA". The name "Africa" should be "The Republic of South Africa". However, BIOVAC is the name of the company and Gyrovac is the name of the commercial vaccine product. The above-mentioned revised parts are in Table 1 and are displayed in red font.

Comments 12: Line 131: The spell of “propionolactone” should be propiolactone.

Response 12: Thank you for pointing this out. I agree with this comment. The revised parts are in line 183 and are displayed in red font.

Comments 13: Line 196: NDV should be Newcastle Disease Virus (NDV)

Response 13: Thank you for pointing this out. I agree with this comment. The revised parts are in lines 247 - 248 and are marked in red font.

Comments 14: Line 202: MDV should be Marek’s Disease Virus (MDV).

Response 14: Thank you for pointing this out. I agree with this comment. But I mentioned "MDV should be Marek’s Disease Virus (MDV)" in line 105 of the revised manuscript, so this part remains MDV.

Comments 15: Line 222: Marek’s Disease Virus (MDV) should be MD.

Response 15: Thank you for pointing this out. I agree with this comment. The revised part is in line 273 and is displayed in red font.

Comments 16: Lines 309-429: The style of references does not meet the journal style. The author's name does not need to be capitalized. All authors should be listed. Please do not use “et al.”. Please delete [j] from the reference list. Please do not omit page rang

Response 16: Thank you for pointing this out. I agree with this comment. Therefore, I will carefully revise the format of the references as required and display it in blue font.

Round 2

Reviewer 1 Report

Comments and Suggestions for Authors

Line 38: the word " toxins"not appropriate  " could be replaced by "virus shedding

Line 90 " toxins" not appropriate could be replaced by "virus shedding" 

table 1 : Gyrovac BIovac " South Africa not correct" should replaced by "Israel" 

Reviewer 2 Report

Comments and Suggestions for Authors

The manuscript (vaccines-3426219) entitled “Advances in CIA Vaccines” by Haoqian Wang et al. has been reconstructed as the Review and revised according to the suggestions and comments of reviewers. This reviewer thinks the authors’ revision solved my major issues and that the manuscript describes the chicken infectious anemia virus vaccines more clearly than before.

However, this manuscript still has minor issues that should be cleared before accepting for publication.

Minor comments.

  1. Line 2, Title: The full name should be used in the title rather than the abbreviated name. Since an exceptional is the gene or protein name, CIA should be Chicken Infectious Anemia. Regarding the “CIA vaccine”, “CIA vaccine” and “CIAV vaccine” is used regardless of discrimination in the main text. Please use it uniformly throughout the manuscript.
  2. Lines 7-8, Abstract: A sentence, “Chicken infectious anemia is a globally distributed immunosuppressive disease, resulting in substantial economic losses for the poultry industry.”, had better be described as “Chicken infectious anemia (CIA) is caused by CIA virus (CIAV) and is a globally distributed immunosuppressive disease, resulting in substantial economic losses for the poultry industry.”
  3. Lines 38 and 90: Please clarify the meaning of “toxins”. The text does not explain any toxins related to CIAV, nor is it shown in the references. Please show the evidence.
  4. Lines 56-57, 202, and 210: the genus name, family name, and species name should be italicized.
  5. Line 60: The “body” had better be shown as the “host”.
  6. Line 61: A non-structural “one” had better be shown as a non-structural “protein”.
  7. Line 75: The “VP3 protein produced” had better be shown only “VP3”.
  8. Line 126: Please insert a period between “(Figure 1)” and “It”.
  9. Line 126: Please clarify the meaning of “literature reports”. Do they mean national guidelines, national requirements, or national written standards? 
  10. Lines 138-139: Please insert “multiplicity and” just before “antigenicity.”
  11. Lines 152-153 and 179: The word “chickens” had better be changed to “hens”, because the authors describe their progeny.
  12. Line 171: Please clarify the meaning of continuous passages. Do they mean the horizontal passages during the chickens in the poultry house? 
  13. Table 1: Please delete “The” from “The Republic of South Africa”.
  14. Line 173: Please insert “Killed” between “3.2. Inactivated” and “Vaccines” to give a contrast with “3.1. Attenuated Live Vaccine”.
  15. Line 175: Please insert “multiplication” before “activity”.
  16. Line 189-190: The sentence is unclear. Please clarify the meaning of “certain limitation” of the protection immunity induced by the inactivated vaccines.
  17. Lines 202, 217, 233, 247, 253, 269, and 282: Several sentences start from “Some researchers” or “Some scholars”. This reviewer feels strange about their use. Why do you not show the name of the first author of the reference or otherwise show the sentence in passive voice?
  18. Line 203: The term “rPiINF-gamma“ had better be shown as “recombinant pigeon INF-gamma (rPiINF-gamma)”.
  19. Line 206: Please insert “CIAV” between “inactivated” and “vaccine”.
  20. Lines 215-216: A sentence is not clear. Do the authors mean “the CpG-ODN can better fortify the cellular and humoral immunity of the CIAV subunit vaccine and achieve a superior immune response outcome than Freund's adjuvants”? Please clarify the meaning of “better than what”.
  21. Line 218: The term “chIL-12“ had better be shown as “Chicken IL-12 (chIL-12)”.
  22. Line 222: The meaning of “the disadvantages of the CIAV attenuated live vaccine” is unclear. Do the authors mean, “Limited application for ages of 9 to 15 weeks is the disadvantage of the CIAV attenuated live vaccine”? Please clarify the meaning of the disadvantages.
  23. Line 235: The words “the 01-4201 strain” had better be shown as “the CIAV strain (01-420)” corresponding to CIAV strain (Cux-1), line 146, CIAV strains (CI 34 and CRI 18), line 147.
  24. Lines 247-249: Please show the immunizing age (days old) with recombinant Newcastle Disease Virus vaccine.
  25. Line 278: Please confirm if or not VP1 had better be changed to VP2-VP1. 
  26. Lines 284 and 287: HMGB1 is defined as a high mobility group box 1 protein categorized as a non-histone protein, but “deltaC” is not. Does it mean the carboxy-terminally truncated form of HMGB1? Please clarify the meaning of “deltaC”.
  27.  Line 299: The words “preventing CIAV” are not clear. Does it mean preventing the infection of CIAV or the onset of CIA? Please clarify the meaning.
  28. Lines 308-309: The meaning of a sentence is unclear. Does this sentence mean, “If the viral load of CIAV could be significantly enhanced enough for industrial applications, it would be used to produce the inactivated vaccine and play a crucial role in the prevention and control of CIAV”? Please clarify the sentence.
  29. Line 332: Please clarify the meaning of “virus’ variation patterns”. Does it mean the “ranges of variation on antigenic epitopes inducing the virus neutralizing antibodies”? Please clarify the meaning.
  30. Lines 376-378: Please clarify what APC is. In addition, please clarify from where the authors get the grant (30801561). 
Comments on the Quality of English Language

Some English related issues are included in the comments and suggestions files for authors.

I am not the native English speaker but feel strange in some parts.

Several sentences start from “Some researchers” or “Some scholars”. I feel strange about their use. I think  the authors  had better  show the name of the first author of the reference or otherwise show the sentence in passive voice.
